

# Comparative diagnostic accuracy of multiparametric magnetic resonance imaging-ultrasound fusion-guided biopsy *versus* systematic biopsy for clinically significant prostate cancer

Jian-hua Fang[1], Liqing Zhang[2], Xi Xie[3], Pan Zhao[4], Lingyun Bao[1] and Fanlei Kong[1]

[1] Department of Medical Ultrasound, Affiliated Hangzhou First People's Hospital, Zhejiang University School of Medicine, Hang Zhou, Zhejiang, China
[2] Department of Radiology, Affiliated Hangzhou First People's Hospital, Zhejiang University School of Medicine, Hang Zhou, Zhejiang, China
[3] Department of Urology Surgery, Affiliated Hangzhou First People's Hospital, Zhejiang University School of Medicine, Hang Zhou, Zhejiang, China
[4] Department of Pathology, Affiliated Hangzhou First People's Hospital, Zhejiang University School of Medicine, Hang Zhou, Zhejiang, China

Corresponding author
Fanlei Kong, kongfanlei@126.com

## ABSTRACT

**Purpose:** To examine the accuracy of transperineal magnetic resonance imaging (MRI)-ultrasound (US) fusion biopsy (FB) in identifying men with prostate cancer (PCa) that has reached a clinically relevant stage.

**Methods:** This investigation enrolled 459 males. In 210 of these patients (FB group), transperineal MRI/US fusion-guided biopsies were performed on the suspicious region, and in 249 others, a systematic biopsy (SB) was performed (SB group). We compared these groups using Gleason scores and rates of cancer detection.

**Results:** PCa cases counted 198/459 (43.1%), including 94/249 (37.8%) in the SB group and 104/210 (49.5%) in the FB group. FB was associated with higher overall diagnostic accuracy relative to SB (88.5% and 72.3%, $P = 0.024$). FB exhibited greater sensitivity than SB (88.9% and 71.2%, $P = 0.025$). The area under the curve for FB and SB approaches was 0.837 and 0.737, respectively, such that FB was associated with an 11.9% increase in accuracy as determined based upon these AUC values. Relative to SB, FB was better able to detect high-grade tumors (GS ≥ 7) (78.85% *vs.* 60.64%, $P = 0.025$).

**Conclusion:** Transperineal MRI-US fusion targeted biopsy is superior to the systematic one as an approach to diagnosing clinically significant PCa, as it is a viable technical approach to prostate biopsy.

## INTRODUCTION

Prostate cancer (PCa) has become the second most common cancer among men worldwide, with incidence rates steadily rising in recent years (*Culp et al., 2020*;

*Sung et al., 2021*). Approximately 1,414,259 new cases and 375,304 deaths were reported globally in 2020 (*National Health Commission of the People's Republic of China, 2022*). Serum prostate-specific antigen (PSA) is commonly used for the initial screening of prostate cancer. However, due to the low specificity of PSA, overdiagnosis and overtreatment have become a cause for concern. In addition, other conditions such as benign prostatic hyperplasia (BPH) or prior urological surgery may lead to elevated PSA levels. Hence, a number of PSA derivatives, including PSA density, velocity, doubling time, and free/total ratio have emerged to improve PCa risk stratification. More recently, tests like the Prostate Health Index (PHI) and the Prostate Cancer Gene 3 (PCA3) have been developed to further refine diagnosis. Multiparametric magnetic resonance imaging (mp-MRI) has now become integral for prostate cancer detection by improving identification of clinically significant tumors, guiding biopsy site seletion, and reducing unnecessary sampling in patients with normal scans. However, trans-rectal ultrasound (TRUS)-guided biopsy remains the current "gold standard" for the diagnosis of prostate tumors (*Park et al., 2015*).While imaging-guided biopsy is common for many solid tumor types, PCa is traditionally detected *via* TRUS-guided systematic biopsy, which consists of random sampling of the entire prostate (*Filson et al., 2016*). However, the current standard 12-core systematic biopsy (SB) approach has been correlated with a 40% false-negative rate, leading the mis-or under-diagnosis of tumors in 35% of assessed individuals (*Ukimura et al., 2013*; *Ma et al., 2018*). Indeed, such SB procedures may fail to properly identify intermediate and high-risk PCa, leading to underestimating the tumor Gleason score (GS) values (*Cowan et al., 2020*; *Mischinger et al., 2022*). Such misdiagnoses can adversely impact patient prognosis by interfering with their treatment planning (*Sönmez et al., 2019*). Notably, distinguishing between clinically significant (csPCa) and non-clinically significant prostate cancers remains a challenge in cancer screening. Currently validated methods for localizing and characterizing prostate tumors mainly rely on the pathologic Gleason score. CsPCa is defined by a biopsy GS of 7 or higher, with increasing aggressiveness as the score rises to 8, 9 and 10. This demonstrates that the Gleason grading system is a key tool for determining the clinical significance and prognostic outlook of prostate cancers (*Dominguez et al., 2023*).

Multiparametric magnetic resonance imaging has proven particularly effective in this setting due to recent advancements in tumor imaging modalities that have allowed for more exact localization and visualization of suspicious lesions. There is substantial evidence that mp-MRI followed by MRI-targeted biopsy (MR-FB) increases the success rate of detecting csPCa while decreasing the success rate of detecting insignificant PCa (insignPCa) (*Hou et al., 2022*; *Majchrzak et al., 2021*; *Tavakoli et al., 2023*). However, MR-FB approaches require access to specialized equipment and experienced operators, making their widespread implementation difficult (*Hanna et al., 2019*).

A number of different prostate biopsy approaches have been developed following the introduction of mp-MRI, such as software-assisted MRI-TRUS fusion biopsy (fus-MRTB), cognitive targeted biopsy (cog-MRTB), and direct in-bore MRI targeted biopsy (inbore-MRTB) (*Bass et al., 2022*; *Lee et al., 2020*; *Wegelin et al., 2017*; *de Ven WJ et al., 2016*). All of

these methods depend on interpreting mp-MRI results as coordinates for directing biopsy needles to suspicious regions of interest (ROIs). The cog-MRTB and fus-MRTB approaches in particular have gained more widespread use as they allow for lower cost and more efficient tumor detection (*Campa et al., 2019*; *Westhoff et al., 2019*). The fus-MRTB approach has also been shown to be more accurate and more reliable than the cog-MRTB approach in some studies. There are two primary forms of fus-MRTB: MRI-TRUS-guided fusion transperineal biopsy (fus-MRTPB) and MRI-TRUS fusion-guided trans-rectal biopsy (fus-MRTRB) (*Goldenberg, Nir & Salcudean, 2019*; *Costa et al., 2015*). However, there is limited data on the diagnostic performance of fus-MRTPB for csPCa. Therefore, the current study aimed to assess and compare the diagnostic efficacy of fus-MRTPB *vs* systematic biopsy for detecting csPCa.

## METHOD

The study has been approved by the Ethics Committee at the First People's Hospital of Hangzhou, affiliated with Zhejiang University School of Medicine (No: 2021-006-01), and conducted in accordance with the Helsinki Declaration. All participants signed informed consent forms.

### Patients

Prostate biopsies were done on 480 individuals with prostate lesions from Affiliated Hangzhou First People's Hospital Zhejiang University School of Medicine between June 2021 and May 2022. Reasons to do a biopsy included: (1) PSA levels that are high (more than 4 ng/ml), (2) visible hypo-echoic lesions upon TRUS, or (3) a positive digital rectal exam. Patients were then randomly allocated in a 1:1 ratio to receive either systematic biopsy (SB group) or fus-MRTPB (FB group). A total of 21 patients were excluded from this study for incomplete information, leading to the final enrollment of 459 patients, of whom 210 underwent mp-MRI scans within 6 months and were subjected to fus-MRTPB (FB group), and of whom 249 that did not undergo mp-MRI scans underwent systematic 12-core TRUS-guided trans-perineal biopsy (SB group).

A flowchart of the protocol is given in Fig. 1.

### Pre-procedural examination

A 3.0T system (Magneton Avanto, Siemens Healthcare, Erlangen, Germany) was used to conduct all mp-MRI scans using a pelvic phased-array coil. Signal acquisition was performed with a 6-channel body matrix coil and the system's integrated 12-channel spine matrix coil. The imaging standard used herein was as follows: transverse dynamic contrast-enhanced imaging (DCE), transverse diffusion-weighted images (DWI), transverse and coronal T2-weighted (T2WI) images, and transverse quantitative parameter apparent diffusion coefficient (ADC) (Fig. 2). These scans took 20 min to acquire as a whole, and any problematic lesions were analyzed through the Prostate Imaging Reporting and Data method, Version 1 categorization method, as recommended by the European Society of Urogenital Radiology. All rating was employed by two radiologists with over ten years of expertise in evaluating prostate MRIs. The mp-MRI

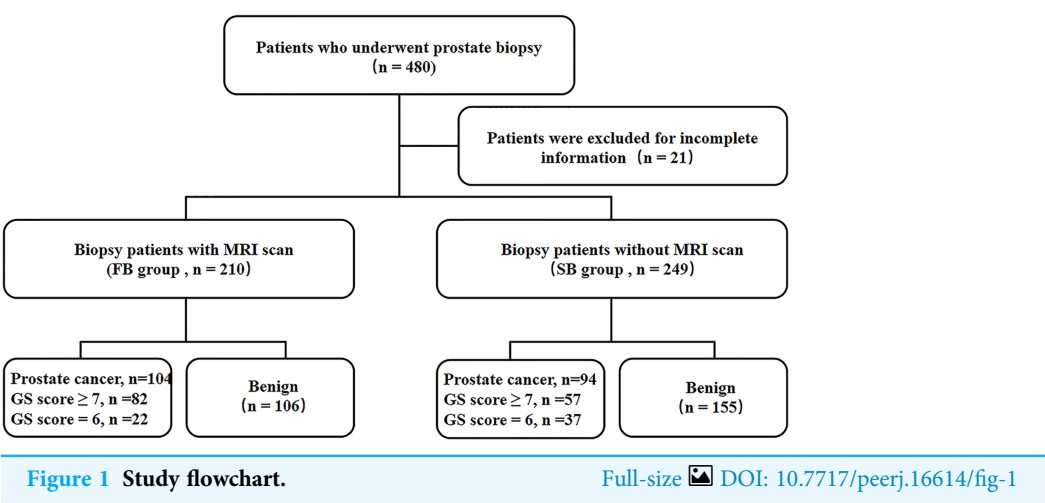

**Figure 1  Study flowchart.**

images were read independently by two radiologists blinded to clinical data. Interobserver and intraobserver agreement for identification of suspicious lesions was assessed using Cohen's kappa statistic.

## US-MRI imaging fusion

A 3.5 MHz convex probe equipped with a needle guiding device and specialized integrated hardware and software (Virtual Navigator, Esaote, Genova, Italy) were used in US scanners (MyLab Twice, Esaote S.p.A., Genova, Italy) for this research. The current investigation made use of the Virtual Navigator MRI-US hybrid framework, which has been given clinical approval by the China Food and Drug Administration. MRI images stored in the institute's database were imported into the fusion system, and contours of the loaded mp-MRI were then drawn and computed to generate a 3D model. Real-time TRUS was then superimposed, allowing for visualization of the ESAOTE Mylab Twice color Doppler US system with the Virtual Navigator system, at a probe frequency of 3–8 MHz. Patients were instructed to adopt the lithotomy position, and trans-perineal biopsy was performed. The ESAOTE Fusion imaging software was utilized to introduce the DICOM format mp-MRI raw data into the RVS ultrasound host. The magnetic field generator is put on one side of the subject's body. The magnetic field position sensor used for TRUS image acquisition is connected to the US probe. During fusion, the prostate gland is first outlined in axial T2WI, DWI, or DCE images, setting the prostate base, apex, and urethral orifice. The resulting 3D model of the gland allows the user to manually define regions that correspond to the MRI-identified lesions of interest (*Cattarino et al., 2019*). The reconstructed fused image is then displayed on the US machine, enabling spatial position-free tracking (Fig. 3). During each fusion-guided biopsy procedure, the operator first localized any suspicious lesions identified on MRI. The radiologist then re-verified the marked lesion to confirm concordance between the operator and radiologist in identifying biopsy targets. Standardized mp-MRI acquisition protocols were utilized for all cases, with image interpretation performed by the same experienced radiologist (>10 years) to ensure

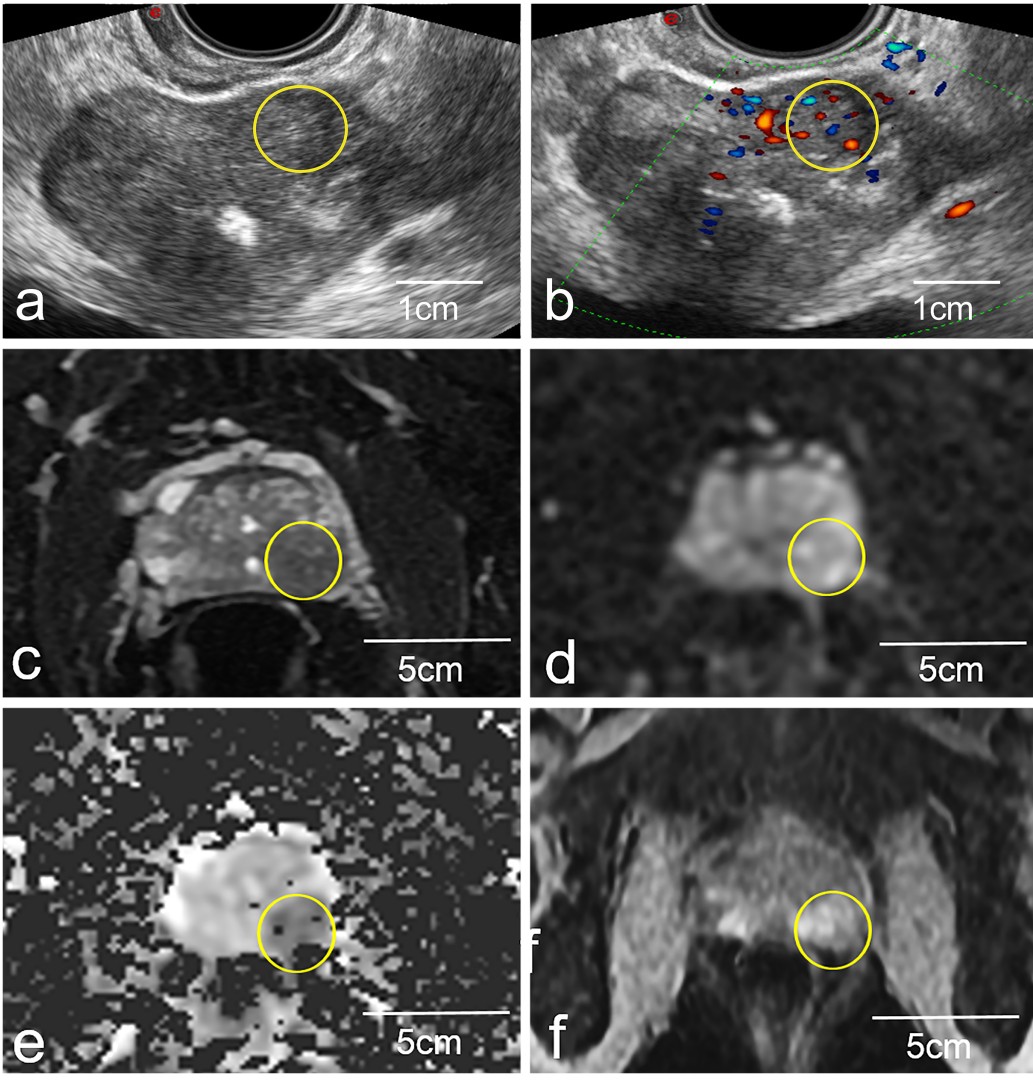

**Figure 2 Representative images of a biopsy patient obtained *via* US (A and B) and mp-MRI (C–F).**
The yellow circled area on the mp-MRI images (C–F) corresponds to the same anatomical location as the yellow circled area on the ultrasound images (A and B). Yellow circles (A and B) highlight a local non-uniform echoic region in the left peripheral zone. This same lesion was more apparent in standard 2D T2W images (C), and high signal upon DWI evaluation (D), with low signal upon ADC evaluation (E) and was strengthened in DCE images (F).

diagnostic consistency. This enabled reliable characterization of lesion location, size, and features for real-time precise targeting under TRUS/MRI fusion guidance.

## Biopsy procedures

Each subject had a prebiopsy enema the morning of the procedure. Authors instructed participants to avoid aspirin and other non-steroidal anti-inflammatory drugs for a week before the biopsy. While in the lithotomy position, a needle guiding device was used for both the SB and FB approaches. All biopsies were conducted by the same doctor at the same time. Suburethral blockade was performed with 1% lidocaine administered into the dorsal prostatic capsule for local anesthetic. Each core was taken at a distinct location,

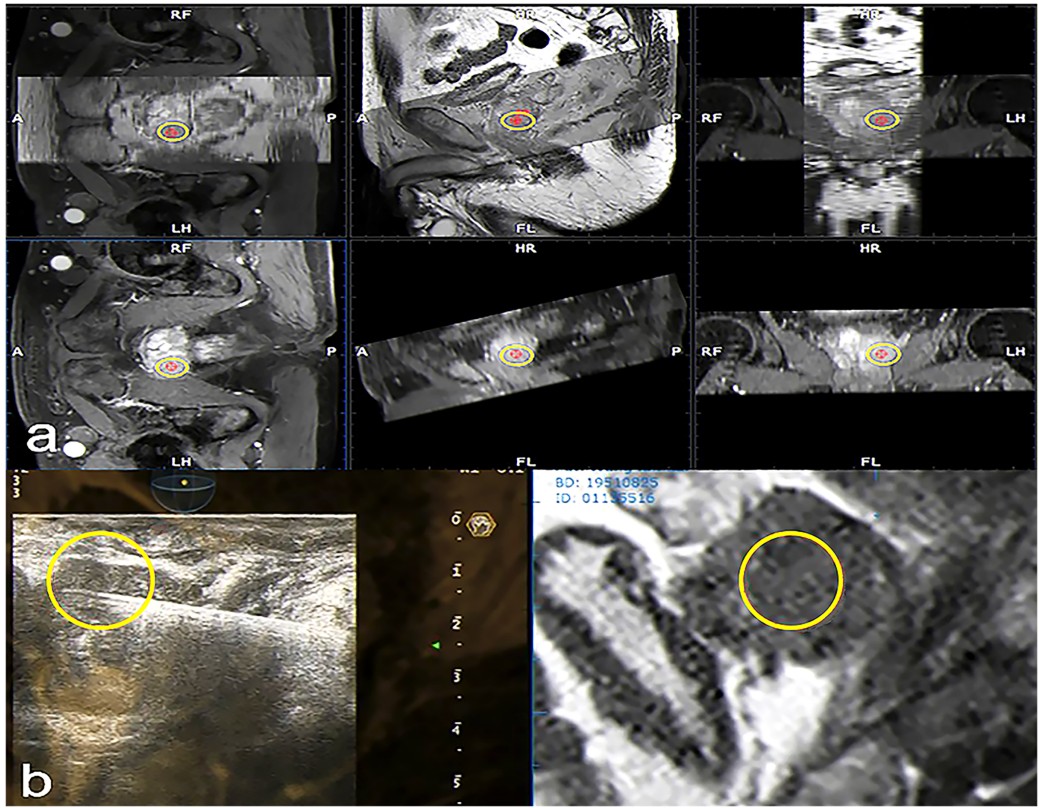

**Figure 3 Fused MRI and trans-rectal US images for a patient with suspected PCa used to guide biopsy procedures.** In the reconstructed sagittal cross-section of the prostate reconstructed from MRI data, the suspicious lesion is highlighted with a yellow circle. Transverse and sagittal MRI sequences showing the suspicious lesion are imported into the ultrasound system. The system automatically fuses the two sequences through the reconstructed MRI image. Subsequently, a target (yellow circle) is marked on the transverse plane, which is synchronously displayed in the corresponding position on the sagittal plane. (A) The stored MRI data was aligned with manual adjustment under ultrasound guidance. This was then fused with real-time ultrasound imaging, enabling automatic tracking and overlay of the MRI-marked target lesion (yellow circle) onto the prostate ultrasound images during scanning. This allowed for localization of the target lesion (yellow circle) on the prostate ultrasound images (B).

thoroughly recorded, and examined by a trained pathologist. Individual biopsy cores were evaluated by a pathologist, and were identified as tumors and assigned Gleason score values, or were classified as inflammation, benign prostatic tissue, or prostatic intraepithelial neoplasia. Clinically significant prostate cancer was defined as having a GS of 7 or higher in any one biopsy core. While maximum cancer-core length was not included as part of the criteria for clinical significance in this definition. The TRUS-guided biopsy was performed utilizing a biplane probe and an 18 G automated Bard biopsy needle. An organized 12-core biopsy was approached, with cores taken from the periphery (PZ) six times, the transition zone (TZ) four times, and the apex twice (*Jiang et al., 2019*). For fusion biopsy procedures, periodic quality assurance was performed by having a subset of cases evaluated by a second urologist to determine concordance in targeting identified lesions.

Prostate surgery was conducted in the local Department of Urology, and the local Institute of Pathology conducted all histopathological analyses, which were then utilized as diagnostic reference standards. In the present study, biopsy and surgical histopathology served as the diagnostic gold standards for prostate cancer detection. All histopathological examinations were conducted by one of six pathologists, blinded to MRI and ultrasound findings, with 5–15 years of experience in prostate pathology. Final Gleason score assignments were based on pathological diagnosis from radical prostatectomy specimens when available, representing the highest standard of truth.

## Statistical analysis

The McNemar test or a generalized estimating equation test for matched data were employed for comparing FB and SB diagnostic parameters. As mentioned by *DeLong, DeLong & Clarke-Pearson (1988)* before, we compared the FB and SB methods by calculating the area under the ROC curve (AUC) for each. The diagnostic efficacy of these two methods was compared using chi-square testing. Using R v.2.3.0, we determined the sensitivity, specificity, positive and negative predictive values (PPV and NPV), and 95% confidence intervals (CIs). Data normality was estimated through the Kolmogorov-Smirnov test and means SD and Student's t-tests were used for normally distributed data, while medians interquartile ranges and Mann-Whitney U tests were used for non-parametric data. Statistical analyses were performed using SPSS v.24.0 (IBM, NY, USA) unless otherwise specified. $P < 0.05$ indicates significance.

# RESULTS

## Patient characteristics

From June 2021 through May 2022, there were 459 participants in this investigation in this hospital, with 249 participants being assigned to the SB group and 210 to the FB group. Baseline demographic and clinical characteristics, along with biopsy procedural details for TB and SB groups are presented in Table 1. No significant differences were observed between the two groups (all $P > 0.05$).

## Comparison of the biopsy parameters for the SB and FB groups

PCa was diagnosed in 198 of 459 total patients (43.1%), including 94/249 (37.8%) in the SB group and 104/210 (49.5%) in the FB group. PCa detection rates were higher in patients that underwent FB relative to those that underwent SB in a by-patient analysis (McNemar's test $P = 0.04$). As seen in Table 2, the average time it took to perform the biopsy was considerably lower in the FB (21.82 min.) than the SB group (23.19 min.) ($P < 0.05$). Gleason score values for all 82 tumors detected in the FB group were ≧7, whereas all 57 tumors detected in the SB group had scores of 7 or higher. Overall, 78.9% (82/104) of PCa lesions with a Gleason score ≥ 7 were detected *via* fusion-targeted biopsy, while just 60.6% (57/94) were detected in the SB group (Table 2). Concerning the consistency between histological methods for identifying PCa, SB and FB were in concordance with the prostatectomy specimen in 64.8% (61/94) and 75.0% (78/104), respectively (Table 3). The interobserver agreement of the two methods was excellent for
**Table 1  Patient characteristics of SB group and FB group.**

| Parameters | SB group | FB group | P vlaue |
|---|---|---|---|
| Patients (n) | 249 | 210 | |
| Body mass index, kg/m$^2$ (IQR) | 24.62 (22.90–27.61) | 25.31 (23.63–28.75) | 0.125 |
| Average age, yr (IQR) | 72 (54–82) | 75 (58–88) | 0.176 |
| Median prostate volume, cm$^3$ (IQR) | 58 (54–76) | 54 (51–83) | 0.710 |
| Serum tPSA, ng/ml (IQR) | 12.21 (5.16–28.08) | 11.24 (5.37–24.90) | 0.844 |
| Mean f/t PSA ratio (SD) | 0.16 (0.07) | 0.17 (0.07) | 0.826 |
| Family history of prostate cancer, n. (%) | 39 (16) | 32 (15) | 0.900 |
| Positive findings in DRE, n. (%) | 17 (7) | 13 (6) | 0.932 |
| Patients with previous negative biopsy, n. (%) | 4 (2) | 2 (1) | 0.839 |
| MRI region of interest diameter, mm; mean (SD) | – | 11 (6) | – |

Note:
SB, systematic biopsy; FB, fusion-guided biopsy; IQR, interquartile range; PSA, prostate specific antigen; SD, standard deviation; DRE, digital rectal examination.

**Table 2  Comparison of the parameters between two groups.**

| Parameters | SB group (n = 249) No. (%) | FB group (n = 210) No. (%) | P value |
|---|---|---|---|
| Overall detection of PCa | 94 (37.75) | 104 (49.52) | 0.04 |
| Age (years) | | | 0.68 |
| <65 | 6 (6.38) | 9 (8.65) | |
| 65~80 | 72 (76.60) | 81 (77.89) | |
| >80 | 16 (17.02) | 14 (13.46) | |
| Prostate volume | | | 0.80 |
| ≤50ml | 39 (41.49) | 45 (43.27) | |
| >50ml | 55 (58.51) | 59 (56.73) | |
| Level of PSA | | | 0.75 |
| <10 ng/ml | 33 (35.11) | 40 (38.46) | |
| ≥10 to <20 ng/ml | 46 (48.94) | 51 (49.04) | |
| ≥20 ng/ml | 15 (15.96) | 13 (12.50) | |
| Gleason score | | | 0.01 |
| GS ≥ 7 | 57 (60.64) | 82 (78.85) | |
| GS = 6 | 37 (39.36) | 22 (21.15) | |
| Mean operation time, min. (SD) | 23.19 (5.63) | 21.82 (3.95) | 0.03 |

Note:
SB, systematic biopsy; FB, fusion-guided biopsy; PCa, prostate cancer; GS, gleason score.

the identification of suspicious lesions, with a kappa value of 0.85 ($P < 0.001$) for FB and 0.88 ($P < 0.001$) for SB. The intraobserver agreement of the two methods was also excellent for identifying suspicious lesions, with a kappa value of 0.81 ($P < 0.001$) for FB and 0.84 ($P < 0.001$) for SB. In the quality assurance analysis, the concordance between urologists in fusion biopsy targeting was 95% based on 20 randomly selected cases.

**Table 3 Gleason score in the combination of fusion-guided and systematic biopsy compared to final histopathology.** Concordance of Gleason score in combination PBx and prostatectomy specimen is marked with bold.

| Parameters | Prostatectomy specimen | | | | | |
|---|---|---|---|---|---|---|
| | GS 6 | GS 3 + 4 | GS 4 + 3 | GS 4 + 4 | GS > 8 | Total |
| SB | | | | | | |
| GS 6 | **16** | 15 | 4 | 2 | 0 | 37 |
| GS 3 + 4 | 2 | **11** | 8 | 1 | 0 | 22 |
| GS 4 + 3 | 0 | 3 | 9 | 2 | 2 | 16 |
| GS 4 + 4 | 0 | 1 | 1 | **11** | 0 | 13 |
| GS > 8 | 0 | 0 | 2 | 1 | **3** | 6 |
| Total | 18 | 30 | 24 | 17 | 5 | 94 |
| FB | | | | | | |
| GS 6 | **12** | 2 | 6 | 2 | 0 | 22 |
| GS 3 + 4 | 2 | **10** | 6 | 0 | 0 | 18 |
| GS 4 + 3 | 0 | 4 | **18** | 4 | 0 | 26 |
| GS 4 + 4 | 0 | 0 | 2 | **16** | 2 | 20 |
| GS > 8 | 0 | 0 | 2 | 4 | **12** | 18 |
| Total | 14 | 16 | 34 | 26 | 14 | 104 |

Note:

    PBx, prostate biopsy; GS, gleason score; SB, systematic biopsy; FB, fusion-guided biopsy.

**Table 4 Diagnostic index of SB and FB for prediction of final prostatectomy specimen.**

| Parameters | SB group | FB group | *P* value |
|---|---|---|---|
| Sensitivity | 0.712 (0.603–0.822) | 0.889 (0.800–0.978) | 0.025 |
| Specificity | 0.762 (0.571–0.905) | 0.857 (0.571–1.000) | 0.595 |
| PPV | 0.912 (0.825–0.982) | 0.976 (0.927–1.000) | 0.388 |
| NPV | 0.432 (0.270–0.595) | 0.545 (0.273–0.818) | 0.509 |
| Accuracy | 0.723 (0.628–0.809) | 0.885 (0.788–0.962) | 0.024 |
| AUC | 0.737 (0.615–0.859) | 0.873 (0.713–1.000) | 0.147 |

Note:

    SB, systematic biopsy; FB, fusion-guided biopsy; PPV, positive predictive value; NPV, negative predictive value. Numbers in parentheses are 95% CIs.

## Evaluation of the relative performance of FB and SB approaches

The FB approach was associated with greater accuracy than the SB approach when using the final pathology findings as reference standards for analysis (88.5% *vs*. 72.3%, $P = 0.024$). The SB approach exhibited respective specificity and sensitivity values of 76.2% (95% CI [0.571–0.905]) and 71.2% (95% CI [0.603–0.822]), as well as NPV and PPV values of 43.2% (95% CI [0.270–0.595]) and 91.2% (95% CI [0.825–0.982]), respectively. In contrast, these values for FB were 88.9% (95% CI [.800–0.978]), 85.7% (95% CI [0.571–1.000]), 97.6% (95% CI [0.927–1.000]), and 54.5% (95% CI [0.273–0.818]), respectively. FB was thus correlated with greater sensitivity relative to SB ($P = 0.025$), while specificity, PPV, and NPV were similar among both groups ($P > 0.05$; Table 4). AUC values

were used to assess the accuracy of these two approaches, revealing FB (AUC = 0.837) to be 11.9% more accurate than SB (AUC = 0.737).

## DISCUSSION

The ideal prostate biopsy approach is one wherein both the lesions and the biopsy process can be readily visualized. In this regard, mp-MRI offers clear advantages over other imaging modalities as a tool for lesion visualization. However, MRI-directed targeting is a complex procedure that is difficult to implement on a large scale. In contrast, US offers clear advantages of allowing clinicians to readily monitor the entire biopsy procedure and biopsy path. MRI-US fusion-targeted biopsy can combine the benefits of these two imaging approaches to improve overall biopsy efficacy.

Recent studies have compared targeted biopsy using MRI/US fusion to systematic biopsy procedures for determining whether prostate lesions are benign or cancerous. For instance, *Bey et al. (2018)* analyzed 332 patients and found that PCa was present in 65 (57.0%) and 70 (59.3%) of the MR-PB and US-PB groups, respectively. However, the MR-PB group had an increased total detection rate (*Bey et al., 2018*). On a per-core basis, targeted biopsy approaches are more efficient than standard prostate biopsy techniques when diagnosing PCa, and this is particularly true for high-grade PCa (*Cattarino et al., 2019*). In a separate study, Jae et al. detected cancer in 23.8% (243/1,021) of cases in their US-PB cohort and 31.3% (399/1,179) of cases in the other MRI-PB cohort, with 22.0% and 31.7% of patients in these two respective groups exhibiting clinically significant PCa, and with MRI-PB exhibiting superior overall cancer detection rates, particularly in the context of clinically significant disease (*Fütterer et al., 2015*). We also observed clear differences between the SB and FB approaches in terms of sensitivity (71.2% *vs* 88.9%, P = 0.025) and accuracy (72.3% *vs* 88.5%, P = 0.024), whereas specificity (76.2% *vs* 85.7%, P = 0.595), PPV (91.2% *vs* 97.6%, P = 0.388), and NPV (43.2% *vs* 54.5%, P = 0.509) did not differ between these groups. The differences between these two approaches may be attributable to the high sensitivity of mp-MRI as a modality for diagnosing PCa, as it is particularly effective when diagnosing high-risk PCa (*Borkowetz et al., 2016*; *Bae & Kim, 2020*; *Hendriks et al., 2021*). Mp-MRI plays an important role in characterizing PCa lesions. High cellular density (indicated by DWI/ADC sequences), changes in glandular tissue morphology (shown by T2 signal intensity), and neoangiogenesis (shown by DCE) play an important role in PCa lesion characterization. For image-guided biopsy, systematic biopsy is limited in its inability to obtain tissue from a specific lesion since most prostate tumors are not visible on ultrasound. Compared to conventional TRUS biopsies, MRI/US fusion biopsy techniques enable more accurate and reproducible sampling of MRI-visible lesions, improving detection of clinically significant cancers. Anterior and lateral facet lesions in the prostate apex are often missed on initial systematic sampling, while in the FB group, combining accurate mpMRI lesion localization and precise transperineal targeted biopsy improves the accuracy and sensitivity for lesions in this region.

The results revealed a significant difference in total PCa detection rates in the FB and SB groups (49.5% *vs* 37.8%; P = 0.04). Relative to SB, FB was better able to detect high-grade tumors (GS ≥ 7) (78.85% *vs* 60.64%, P = 0.025). Differences in these results may be

attributable to differences in patient populations and reference standards, but this reported outcomes indicate what was previously reported. For example, *Siddiqui et al. (2015)* assessed 1,003 patients and found that MR/TRUS fusion-targeted biopsy did not differ significantly from TRUS-guided systematic biopsy with respect to PCa detection rate, but that this formed approach was able to detect significantly more high-risk tumors relative to the latter approach (173 *vs* 122 cases, $P < 0.001$) while also detecting 17% fewer low-risk tumors (213 *vs* 258 cases, $P < 0.001$) (*Siddiqui et al., 2015*). They determined that their FB strategy superiorly differentiated low from intermediate/high-risk PCa more than a standard biopsy approach or the combination of these two approaches (AUC = 0.73, 0.59, and 0.67, respectively; $P < 0.05$) (*Rouvière et al., 2019*). *Wegelin et al. (2017)* found that MRI-targeted biopsy approaches are valuable as they can improve high-risk PCa detection while reducing the rate of low-risk PCa detection relative to traditional biopsy approaches. The main reason for the improved detection of csPCa by MRI-ultrasound fusion biopsy is the strong correlation between mp-MRI findings and histopathological measures of tumor grade, especially in the peripheral zone. ADC values of suspicious MRI lesions demonstrate stronger association with post-prostatectomy Gleason scores compared to grades from standard TRUS biopsy. This can be attributed to the increased density of prostate cancer cells, altered mesenchymal structure, and fibrosis leading to restricted diffusion of water molecules, which results in decreased ADC values (*Chenevert, Sundgren & Ross, 2006*). Prostate cancer foci appear as low signal on the ADC map, and it is easy to find suspicious foci to place ROI for measurement, and the location of the cancer foci can also be confirmed with the help of radical surgery specimens. In contrast, conventional ultrasound-guided biopsy involves randomized sampling and risks mischaracterizing or entirely missing clinically significant cancer foci, potentially underestimating Gleason grade. Therefore, fusion navigation biopsy with the advantage of MRI can confirm the diagnosis of high-grade PCa, thus increasing the detection rate of csPCa and avoiding delayed treatment.

While one prospective, multicenter, paired diagnostic study observed no differences between FB and SB approaches for detecting ISUP grade group 2 or higher PCa, combining the two modalities improved detection rates. Performing an mp-MRI scan in patients prior to their first biopsy might enhance the rates of csPCa detection, without eliminating the need for conducting a systematic biopsy (*Bae & Kim, 2020*). Such differences may be attributable to three different factors. Firstly, a targeted biopsy is likely to achieve higher cancer detection rates relative to random systematic biopsy. Secondly, these analyses were conducted by two radiologists and multiple urologists with varying levels of experience, potentially impacting the final results. Third, we did not assess possible correlations between pathology and TRUS abnormalities. In this study, a Gleason score ≥ 7 in tumor pathology specimens was used to define csPCa. In fact, some other factors (such as tumor volume, PSA density, and PI-RADS score) are also related to the diagnosis of csPCa. For example, studies have shown that PSA density is associated with the detection of csPCa in radical prostatectomy specimens of small to medium prostate cancers. For patients undergoing evaluation due to elevated PSA and/or asymptomatic prostatic disease, such biomarkers are available and cost-effective tools that can be

incorporated into the decision-making process. Our study mainly assessed the diagnostic performance of the two biopsy methods through the pathology of biopsy specimens, without incorporating the above clinical indicators that affect csPCa (Omri et al., 2020).

There are multiple limitations to this study. First, targeted puncture efforts are dependent upon multi-parameter MRI and access to high-performing equipment, and experienced operators. Second, the utility of the MRI/TRUS fusion technology can vary in certain situations, and is currently not well-suited to the accurate localization of lesions <3 mm in size. Third, this was a single-center study and is thus susceptible to potential selection bias. Finally, we only compared MRI/TRUS fusion biopsy to systematic biopsy, the relative performance of cognitive MRI-targeted biopsy and other emerging modalities warrants further investigation.

## CONCLUSION

In summary, trans-perineal MRI-ultrasound fusion targeted biopsy is more favorable than 12-cores systematic biopsy as a means of diagnosing PCa associated with a Gleason score ≥ 7, particularly with respect to sensitivity. Future improvements in virtual navigation technology may further extend the utility of trans-perineal MRI-Ultrasound fusion targeted biopsy for patients that need to undergo prostate biopsy, thereby improving clinical outcomes. MRI-Ultrasound fusion targeted biopsy technology has the potential to considerably impact the clinical profiles of men diagnosed with PCa.

The demonstrated higher detection rate of clinically significant tumors and lower rate of insignificant disease could enable more appropriate triage of patients to active surveillance vs treatment. This approach may allow clinicians to better discriminate indolent vs aggressive cancers at the time of diagnosis through enhanced non-invasive characterization of tumor grade and extent. More accurate risk stratification could lead to improved personalized management plans tailored to disease severity from the outset.

### Funding
This study was supported by The Construction Fund of Medical Key Disciplines of Hangzhou (No. oo20200457), and the Science and Technology Development Project of Hangzhou (No. A20200113). The funders had no role in study design, data collection and analysis, decision to publish, or preparation of the manuscript.

### Grant Disclosures
The following grant information was disclosed by the authors:
The Construction Fund of Medical Key Disciplines of Hangzhou: oo20200457.
Science and Technology Development Project of Hangzhou: A20200113.

### Competing Interests
The authors declare that they have no competing interests.

## Author Contributions

- Jian-hua Fang conceived and designed the experiments, authored or reviewed drafts of the article, and approved the final draft.
- Liqing Zhang conceived and designed the experiments, performed the experiments, authored or reviewed drafts of the article, and approved the final draft.
- Xi Xie performed the experiments, authored or reviewed drafts of the article, and approved the final draft.
- Pan Zhao performed the experiments, authored or reviewed drafts of the article, we gratefully acknowledge Dr. Pan Zhao for technical assistance, and approved the final draft.
- Lingyun Bao conceived and designed the experiments, authored or reviewed drafts of the article, and approved the final draft.
- Fanlei Kong conceived and designed the experiments, performed the experiments, analyzed the data, prepared figures and/or tables, authored or reviewed drafts of the article, and approved the final draft.

## Human Ethics

The following information was supplied relating to ethical approvals (*i.e.*, approving body and any reference numbers):

This study was carried out in accordance with the guidelines of the Ethics Committee of the Affiliated Hangzhou First People's Hospital, Zhejiang University School of Medicine, with written informed consent from all subjects (Approval number: No: 2021-006-01). All subjects gave written informed consent in accordance with the World Medical Association Declaration of Helsinki. The protocol was approved by the Affiliated Hangzhou First People's Hospital, Zhejiang University School of Medicine.

## Data Availability

The raw data are available in the Supplemental Files.

## Supplemental Information

Supplemental information for this article can be found online at http://dx.doi.org/10.7717/peerj.16614#supplemental-information.

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
