# Peer review of "Comparative diagnostic accuracy of multiparametric magnetic resonance imaging-ultrasound fusion-guided biopsy versus systematic biopsy for clinically significant prostate cancer"

_PeerJ, doi:10.7717/peerj.16614_

## Round 0.1 · original submission · Major Revisions

Authors should revise according to the suggestions of reviewers. The modifications should be marked. A point-to-point response letter is needed.

Reviewer 1 ·

Basic reporting

1. I value the opportunity to review this interesting manuscript. The aim of the paper entitled “A comparison of the relative performance of trans-perineal MRI-Ultrasound fusion targeted biopsy and systematic biopsy when diagnosing clinically significant prostate cancer” may appropriate for consideration of publication in the PeerJ. However, there are some concern about this manuscript.

2. The data presented in this manuscript is scientifically significant and noteworthy. Notably, the manuscript has been cited in 67.74% (21/31) of current publications within the preceding five years, which include 2 articles in 2018, 8 articles in 2019, 4 articles in 2020, 3 articles in 2021, 3 articles in 2022, and 1 article in 2023.

3. The topic title implies a focus on agreement between different diagnostic methods, whereas the study mainly compares the performance of two methods. A more explicit title and clarification on method selection and limitations would improve reader comprehension while acknowledging the importance of a broader diagnostic accuracy assessment. I recommend you reconsider the article title.

Experimental design

4. As a suggestion, you may consider using a Bayesian model to estimate diagnostic performance. This statistical approach has been shown to be effective in addressing uncertainties in complex diagnostic scenarios; and can provide valuable insights into the performance of diagnostic tests. Therefore, I recommend considering the use of a Bayesian model in your study.

Validity of the findings

5. I have carefully examined your work and would like to suggest a crucial adjustment that may significantly enhance the quality and credibility of your findings. Specifically, it is highly recommended that you refrain from utilizing photographs taken from a camera when presenting images from MRI scans, and instead, export images directly from the system.

6. It is strongly recommended that the author considers the optimization of the field of view (FOV), spatial and contrast resolution, as well as window width and level settings in order to enhance the visual quality of the images and increase the impact of their work. These adjustments have the potential to significantly improve the overall effectiveness of the presented research article.

7. To enhance the precision of measurements and improve the accessibility, reproducibility, and accuracy of research, it is proposed that the inclusion of scale bars in MRI images be considered, given its widespread use in scientific research.

8. Please provide details on the methodology used to establish the gold standard for the diagnosis of prostate cancer, as well as how sensitivity and specificity were calculated in the study. Please also provide additional information on how diagnostic performance was evaluated.

Additional comments

9. Please arrange the keywords alphabetically for a standardized presentation.

10. Ultimately, I express my gratitude for the opportunity to review this submission. While recognizing the significance and appeal of the topic at hand, it is my assessment that additional clarification and impetus is prerequisite. It is my sincere hope that the recommendations offered prove advantageous to the authors in their efforts towards publishing this work.

Reviewer 2 ·

Basic reporting

There are some typos and grammar mistakes in the paper. In general, the writing is okay.

Experimental design

It is unclear to me how the patients were assigned to different groups. Is the waiting time the only determinant factor in this decision? Clarifications from the authors would be appreciated. In addition, the patient characteristics are very limited and a more comprehensive table 1 would provide the necessary information to determine if the patients are comparable between the SB group and FB group.

Validity of the findings

I am not as convinced that the patient populations are comparable between the SB group and FB group. Based on the GS grades, patients in the FB group are associated with more advanced diseases compared to those in the SB group.

Reviewer 3 ·

Basic reporting

This manuscript compares the accuracy of transperineal MRI-US fusion targeted biopsy (FB) and systematic biopsy (SB) for diagnosing clinically significant prostate cancer (csPCa). 459 males are enrolled and underwent either FB or SB. The authors found that FB group had higher overall detection rate of PCa, higher detection rate of high-grade tumors (GSg7), and higher sensitivity and accuracy than SB group, while specificity, PPV, and NPV were similar between the two groups. It concludes that transperineal MRI-US fusion targeted biopsy is more effective than systematic biopsy for diagnosing csPCa and suggests that future enhancements in virtual navigation technology may further improve its utility.

Overall, the manuscript is clear, well-written, and original. The reviewer feels that the paper will be qualified for publication if the author can address the following comments:

1. It is suggested the introduction be expanded to provide more background information on the current state-of-the-art methods for prostate cancer diagnosis, such as blood tests, urine tests, gene tests, or other new diagnostic tests that are available or under development. By contextualizing the research question in this broader context, the significance of FB and SB can be better highlighted.

2. The manuscript includes instances of incomplete or improper formatting. For example, missing blanks for the title of Table 4 “Numbersinparenthesesare95%CIs”.

3. More careful proofreading is required on several details in the manuscript to enhance the clarity of the presentation, i.e. in Line 183, “Pca” should be “PCa”, and in Lines 202 and 207, “for one” could be “firstly” or “for one thing”…

Experimental design

1. The methods section should clarify how the patients were allocated to either FB group or SB group, whether it was random or based on some criteria or preference. This would help to assess the comparability and representativeness of the two groups.

2. Consider performing subgroup analysis or stratification based on potential confounding factors such as age, prostate volume, PSA level, etc. These factors may influence the performance and outcomes of both fusion biopsy and systematic biopsy and accounting for their effects would strengthen the analysis. Adjusting for these factors in your statistical models or performing subgroup analyses will help explain some of the observed differences between the two groups.

Validity of the findings

1. For Figure 3(b), in order to enhance clarity, it is suggested to include a brief explanation of how the target lesion was confirmed as it is not evident from the current picture.

2. The manuscript lacks information regarding the interobserver agreement or reproducibility of the mp-MRI interpretation and fusion biopsy procedures. This information is crucial in evaluating the reliability and validity of these techniques. It would be beneficial to include details on how the consistency and quality of mp-MRI readings and fusion biopsy operations were assessed among different radiologists and urologists.

·

Basic reporting

Authors have made a commendable attempt to improve prostrate cancer diagnosis by developing/improving a novel method that is seemingly more efficient than conventional approaches. However, some major revisions are suggested.

Experimental design

The design seems fine, however, some parts of the results should be in the materials and methods section. By way of example, lines 133-135 have not revealed anything from the finding. Those lines only offer a declaration on the data collection structure. The results section should be revised for this.

Validity of the findings

The discussion only tells about WHAT was found, and not WHY it was found, or HOW the results came about. Other literatures that were cited were just also a declaration of what was reported in their studies. If visualization is the key advantage in the advantage of any diagnostic tool, what are the visualization characteristics that should be sought after? Is an efficient visualization system expected to show the tumors in colored images? or graphs? or some other property?

How is quality defined with regards to prostrate cancer detection using these systems?

Additional comments

Introduction should include an interesting mention of prevalence of prostrate cancer, statistics-wise.

The use of pronouns "our" is not too ethical for a technically written paper.

Conclusion is too brief, and future recommendations seem too short for important subject matter as this.

---

## Round 0.2 · Minor Revisions

Authors should revise according to the suggestions of reviewers. The modifications should be marked. A point to point response letter is needed.

Reviewer 1 ·

Basic reporting

"After reviewing these revisions, it is clear that you have taken my feedback to heart and have made impressive strides in enhancing the manuscript. I am thoroughly impressed by the substantial efforts you have put into addressing the raised concerns. Not only have you revised the research title, but you have also made commendable efforts to enhance the visual quality of the images. I would like to express my utmost appreciation for the insightful discussion on the study limitations and the thorough explanations regarding the absence of certain findings."

Experimental design

None

Validity of the findings

None

Additional comments

None

Reviewer 2 ·

Basic reporting

The authors have addressed all my comments. There are a few minor edits needed.
1. Table 1. It is uncommon to report a mean with 95% CI, usually mean and standard deviation are reported together. Also, serum tPSA, is that mean or median?
2. Table 2: The level of PSA, whether the 10ng/ml is included in the first category or second category is unclear. And please report the SD for mean operation time

Experimental design

No further comments.

Validity of the findings

No further comments.

Additional comments

No further comments.

Reviewer 3 ·

Basic reporting

The manuscript offers a valuable comparative analysis of transperineal MRI-US fusion targeted biopsy (FB) and systematic biopsy (SB) in the detection of prostate cancer. The narrative is largely coherent and the subject matter is pertinent. However, there are several areas that could benefit from further elucidation or revision to enhance the clarity and completeness of the report.

A clear clarification of the figures would be beneficial.
Figure 3: It would be helpful to provide additional captions or a legend (or a scaling bar) to better explain the images presented, especially for readers unfamiliar with Virtual Navigator. Ensuring the images are exported directly from the software rather than photographed would enhance clarity and professionalism.
Figure 2: The non-uniform echoic region’s appearance discrepancy between sub-figures (a) and (b) needs clarification. If the images were captured at different times, this should be stated. Additionally, it should be made clear whether the yellow circles in sub-figures (a-b) and (c-f) correspond to the same region.

The manuscript would greatly benefit from a more thorough discussion of the possible mechanisms underlying the superior performance of FB over SB. Delving into the technical and procedural differences between FB and SB, and how these may contribute to the observed disparities in diagnostic accuracy, sensitivity, and specificity could provide readers with a better understanding of the results.

The manuscript requires thorough proofreading to correct grammatical errors and typos. For instance:
Table 4. Missing FB "group".

Experimental design

The authors should provide a clear definition of clinically significant prostate cancer (csPCa) and justify the use of Gleason Score as the primary criterion. It may be beneficial to discuss or incorporate other factors like tumor volume, PSA density, and PI-RADS score, which have been indicated in other studies to be relevant in csPCa diagnosis.

Validity of the findings

Reporting the intra-observer reproducibility for the FB and SB methods may be helpful for assessing the reliability and validity of the diagnostic findings. These metrics would also provide insight into the potential variability in diagnostic accuracy between different observers or repeated measures by the same observer.

---

## Round 0.3 · accepted · Accept

The authors have addressed the reviewers' concerns properly and revised the manuscript accordingly. The manuscript can be accepted for publication in its current form